# 4D imaging of lithium-batteries using correlative neutron and X-ray tomography with a virtual unrolling technique

Ralf F. Ziesche [1,2], Tobias Arlt[3], Donal P. Finegan [4], Thomas M.M. Heenan [1,5], Alessandro Tengattini[6,7], Daniel Baum [8], Nikolay Kardjilov [9], Henning Markötter[3,9,10], Ingo Manke [9], Winfried Kockelmann[2], Dan J.L. Brett[1,5] & Paul R. Shearing[1,5]*

The temporally and spatially resolved tracking of lithium intercalation and electrode degradation processes are crucial for detecting and understanding performance losses during the operation of lithium-batteries. Here, high-throughput X-ray computed tomography has enabled the identification of mechanical degradation processes in a commercial $Li/MnO_2$ primary battery and the indirect tracking of lithium diffusion; furthermore, complementary neutron computed tomography has identified the direct lithium diffusion process and the electrode wetting by the electrolyte. Virtual electrode unrolling techniques provide a deeper view inside the electrode layers and are used to detect minor fluctuations which are difficult to observe using conventional three dimensional rendering tools. Moreover, the 'unrolling' provides a platform for correlating multi-modal image data which is expected to find wider application in battery science and engineering to study diverse effects e.g. electrode degradation or lithium diffusion blocking during battery cycling.

---

[1] Electrochemical Innovation Lab, Department of Chemical Engineering, University College London, London WC1E 7JE, UK. [2] STFC, Rutherford Appleton Laboratory, ISIS Facility, Harwell OX11 0QX, UK. [3] Technische Universität Berlin, Strasse des 17. Juni 135, 10624 Berlin, Germany. [4] National Renewable Energy Laboratory, 15013 Denver West Parkway, Golden, CO 80401, USA. [5] The Faraday Institution, Quad One, Harwell Science and Innovation Campus, Didcot OX11 0RA, UK. [6] Grenoble INP, CNRS, 3SR, Univ. Grenoble Alpes, 38000 Grenoble, France. [7] Institut Laue-Langevin (ILL), 71 Avenue des Martyrs, 38000 Grenoble, France. [8] Zuse Institute Berlin, Takustraße 7, 14195 Berlin, Germany. [9] Helmholtz-Zentrum Berlin für Materialien und Energie (HZB), Hahn-Meitner-Platz 1, 14109 Berlin, Germany. [10] Present address: Bundesanstalt für Materialforschung und -Prüfung, Unter den Eichen 87, 12205 Berlin, Germany. *email: p.shearing@ucl.ac.uk

In recent years, the advancement of X-ray computed tomography (CT) capabilities have facilitated a broadening of our understanding of battery materials and devices, with studies spanning multiple length scales, from nanometre to millimetre, and multiple time scales from kilohertz to microhertz[1]. These studies have collectively provided insight into the relationship between electrode microstructure and performance[2–4], battery architecture, safety[5–8] and new battery materials[9–11]. While the majority of these studies have utilised X-ray CT, there is growing interest in the application of neutron imaging for battery applications; the complementarities of X-ray and neutron imaging, which are sensitive to electron and nuclear density, respectively, provide significant opportunities for correlative studies. This is particularly true for lithium (Li) batteries, owing to the high neutron sensitivity to low-Z materials (in particular Li and H).

Previous neutron imaging studies on Li cells have shown a high potential for neutron radiography to examine the dynamic Li diffusion process and electrode expansion[12–14], electrolyte consumption[15] and gas evolution[16–18]. Standard neutron tomography has been historically limited in application due to the long exposure times, but enables a three-dimensional static view of the Li distribution[19,20]. To the authors' knowledge, LaManna et al.[21] demonstrated the first use of X-ray and neutron CT as complementary techniques on a Li battery.

The non-destructive nature of both X-ray and neutron CT provides opportunities for four-dimensional (4D) studies, that is, the ability to explore the evolution of three-dimensional (3D) structures with time[22–24]. The authors have previously used high-speed X-ray CT to produce consecutive datasets of batteries during discharge for quantitative displacement measurements via digital volume correlation (DVC)[25]. The DVC of consecutive tomograms was used to explore the evolution of electrode structures in Li primary cells, which are widely used across consumer electronics, backup-power and aerospace applications.

In the current manuscript, we extend our previous study to evaluate the Li distribution in the $MnO_2$ cathode of a commercial CR2 Li-ion primary cell from Duracell[26] using both X-ray and neutron CT. The experiments were performed at the ID15A beamline at the European Synchrotron Radiation Facility (ESRF), known for high-speed and high-resolution imaging capability[6,27], and at the V7 CONRAD-2 neutron imaging beamline at the BER II research reactor at Helmholtz Centre Berlin, known for its combination of high-resolution neutron imaging with step-wise sample evolution[28,29].

The combination of X-ray and neutron CT presented here provides powerful complementary information. The contrast in X-ray imaging is mainly based on the absorption of X-ray photons by their interactions with the electron clouds within the materials. The degree of X-ray absorption tends to increase with the atomic number of the elements; therefore, elements with low atomic numbers, such as Li, display weak interactions and thus low contrast. Neutrons interact with the nucleus of an atom and show high sensitivity for light elements, where the nuclei are more closely packed[30,31]. X-ray CT is leveraged to identify features that are beyond the resolution capability of neutron CT, whereas neutron CT, being highly sensitive to Li, is utilised to track the spatial dynamics of light components such as: the Li metal, Li ions, Li salt and the hydrogen in the electrolyte, as well as formed gases.

To effectively evaluate and compare these multi-modal results, a methodology for virtually 'unrolling' the characteristic spirally wound cell captured in the 3D tomograms is used[32–34]: This provides an effective means for data visualisation and correlation, which, moreover, reveals erstwhile unseen trends in the spatial distribution of performance within these cells. While X-ray CT is increasingly widespread in its application to Li batteries, this

methodology of correlative imaging and data presentation/visualisation will provide a new benchmark for battery characterisation. This can be readily applied to a range of battery geometries and chemistries, providing unique insight into the dynamics of Li transport during charge and discharge.

## Results

**X-ray tomography.** X-ray and neutron imaging were performed to study the Li intercalation, morphology changes and degradation processes in two different commercial CR2 Li/manganese dioxide cells discharged under identical conditions as described in the Methods/Experimental section. In the following sections, the X-ray and neutron imaging analyses are discussed along with the insights achieved from virtual unrolling of the spiral-wound electrode architecture. A schematic illustration of the cell components and operation can be found in Fig. 1a.

Figure 2a shows three of the high-speed X-ray tomograms: one in the pristine state and two at partly discharged state of charge (SoC) after 1500 and 3900 s, where the corresponding SoC and discharge curves are given in Fig. 1b; further details are given in Supplementary Fig. 2a. The tomograms show highly X-ray attenuating battery materials as being bright, such as the nickel current collecting mesh, which is woven in the lower attenuating $MnO_2$ cathode, and the rectangular tab that connects the mesh to the terminal. Owing to the low interaction with X-rays, the Li metal electrode, the separator and the electrolyte are not clearly visible.

These tomograms provide information about the cathode morphology changes and mechanical degradation during discharging. In the pristine state, the cathode already showed cracks, which originated from the production process of the CR2 cell where the electrode-separator layers were wound. Cracks appear mainly at the outer circumference of the winding cathode and more severely in the middle of the cell where the electrode's bending radius is higher, causing an increased strain. During the discharging process, the region where the Li metal electrode is expected to be present diminishes in size. This is caused by Li removal from the anode, while the cathode starts to expand heterogeneously as a result of the Li insertion into the $Li_xMnO_2$ crystal structure.

During the expansion, the electrode showed a slight degree of unravelling, which is caused by the movement of the current collector mesh against the active material as a result of strain triggered by Li insertion. Cracks were observed to expand heterogeneously in the anti-clockwise direction with increasing size into the outer windings, as previously reported by the authors[25]. The current collector tab anchors the electrode at a single point and is the unravelling centre. Stronger $Li_xMnO_2$ cracking is detected in the upper axial cathode section than in the middle section, likely as a result of the weaker electrode compression, which facilitates Li intercalation and electrode expansion (see Fig. 2a, Supplementary Fig. 3 and Supplementary Fig. 4). Delamination of active material from the current collector can create electrically (and therefore electrochemically) isolated areas, which, in turn, can cause power and capacity loss. However, cracks that expose additional $Li_xMnO_2$ surface area, through an increase in the electrode porosity, may also benefit Li insertion and cell performance due to voids that are flooded with highly Li-ion conductive electrolyte.

X-rays are a good probe for detecting mechanical degradation processes in the $Li_xMnO_2$ electrode, such as cracking, delamination or unravelling effects as outlined above, but are unable to observe the electrolyte, the Li removal from the anode or the insertion into the cathode. However, the latter can be inferred by observing the electrode swelling and the loss of image intensity in the $Li_xMnO_2$ electrode.

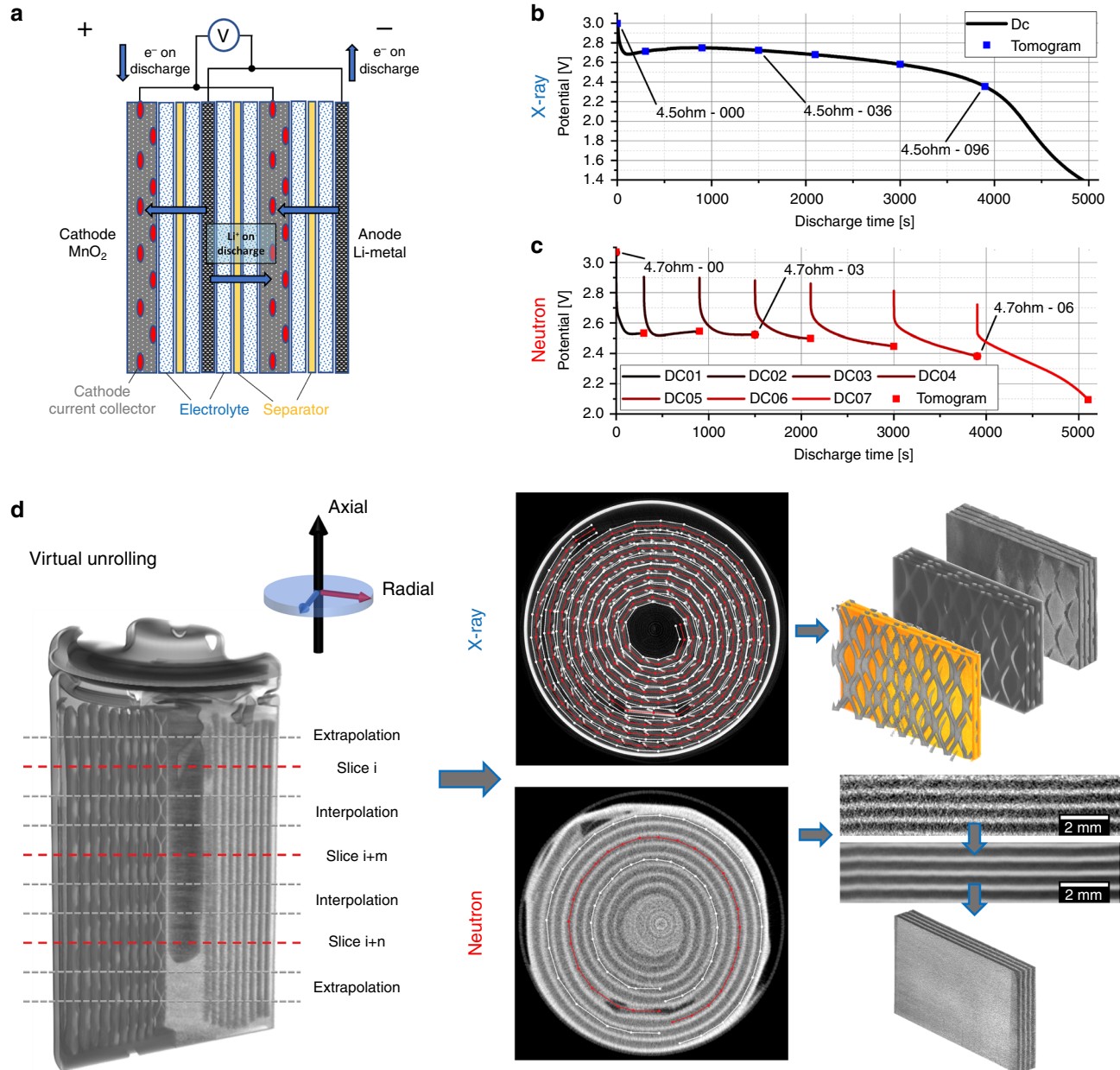

**Fig. 1 Illustration of the cell operation, discharge curves and virtual unrolling technique.** Image **a** shows an illustration of the studied Li/MnO₂ CR2 primary cell from Duracell. Graph **b** shows the constant resistance discharge curve for the CR2 cell over 4.5 Ω, where simultaneous fast X-ray CT was carried out. Graph **c** shows the constant resistance discharge curve over 4.7 Ω, where the discharge was interrupted for each neutron tomography after a certain time interval. Image **d** displays the reconstructed tomograms from neutron and X-ray CT along with examples of sections extracted following virtual unrolling of the reconstructions. Clearly visible in the X-ray images is the nickel current collecting mesh, which appears brighter than the Li$_x$MnO₂ active electrode material.

**Neutron tomography**. Figure 2b shows three neutron tomograms of the CR2 cell at different SoC labelled from CR2-00, the pristine state, to CR2-07. Highly neutron-attenuating elements, such as Li and hydrogen, appear bright in the neutron tomograms, which facilitates detection of the Li anode between the darker, lowly attenuating cathode.

It is observed that an excess of electrolyte filled the middle region of the electrode assembly of the pristine cell. Electrolyte is also observed to occupy the region near the positive terminal at the top of the cell.

Strong mechanical degradation effects on the Li$_x$MnO₂ electrode are clearly visible on the upper axial cathode ending at an early stage of Li insertion. Here, electrode cracking starts

after about 1500 s of discharging at a capacity of −225.71 mAh (CR2-03), whereas in the cell, radial middle and bottom section cracking is less pronounced and limited to the outer radial electrode windings identical to the cracking detected in the X-ray images (see Supplementary Figs. 5 and 6). The strong mechanical degradation in the upper axial section is suspected on a high Li intercalation by the cathode, which can be seen in the top horizontal slices from CR2-03 onwards to CR2-05 of Supplementary Fig. 5. There are regions where the electrolyte is completely removed and the consumed Li between the cathode layers is visible. That strong mechanical effect occurs over a ca. 2 mm high region in the upper axial cell part seen in the vertical slices of Supplementary Fig. 6 from CR2-03 onwards.

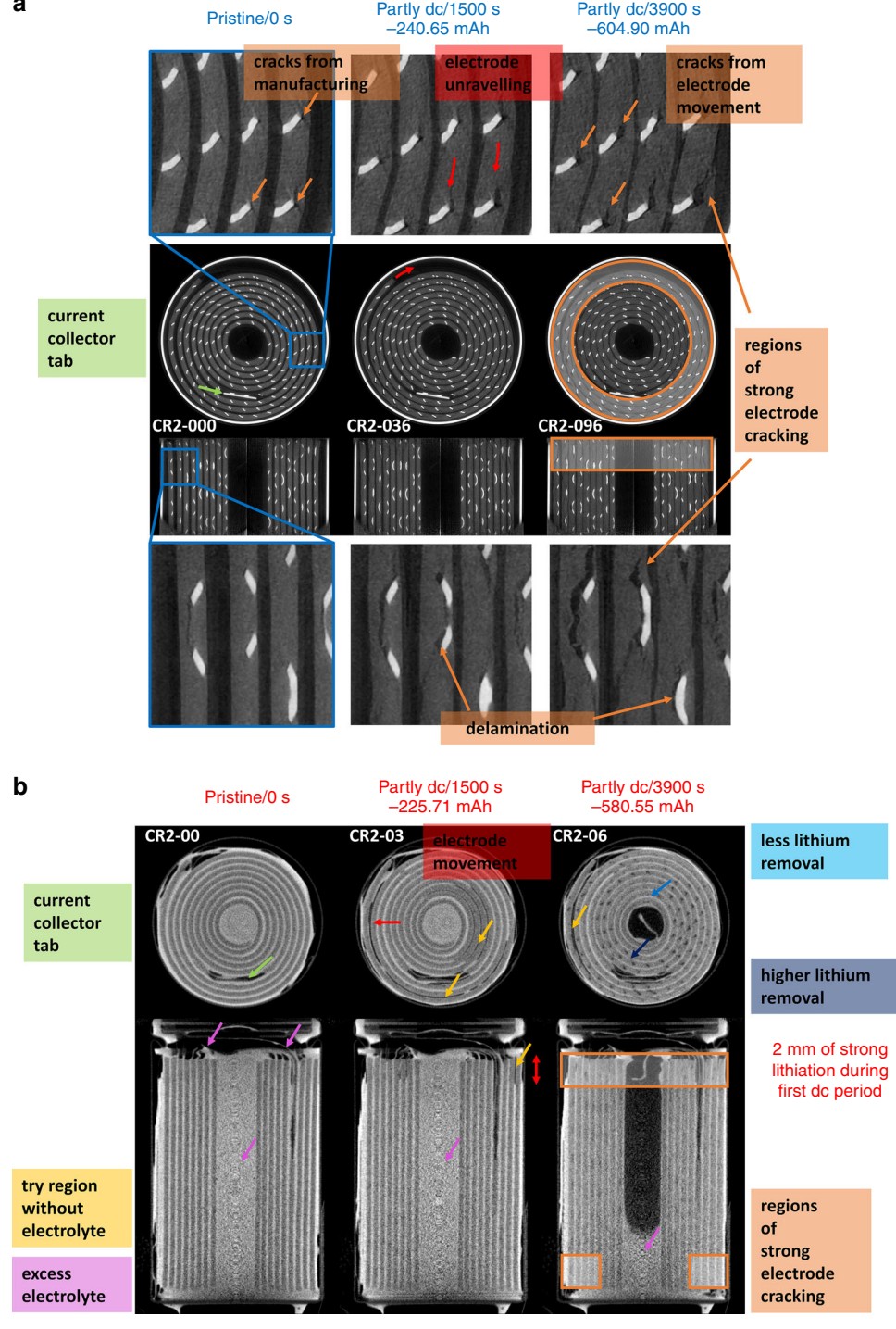

**Fig. 2 3D reconstructed operando X-ray and in situ neutron tomograms. a** shows horizontal and vertical orthogonal slices out of the X-ray tomograms. In total, 103 tomograms were recorded labelled from CR2-000 to CR2-102. One tomogram was recorded every 40 s with a total acquisition period of 2.8 s. Here the pristine state and two partly discharged states are presented. The images show the cracking and volume expansion of the MnO$_2$ electrode during cell discharging. The highly absorbing steel casing is visible as very bright ring around the wounded membrane–electrode ensemble. The contrast was optimised in order to improve the contrast within the lower attenuating components. **b** shows orthogonal slices of the neutron tomogram captured during the discharge over a 4.7 Ω resistor, where the lithium electrode and the excess of electrolyte in the middle of the cell are clearly visible. Lithium intercalation and electrolyte consumption are observed, as well as electrode cracking and electrolyte consumption. In total, eight neutron tomograms were collected with an acquisition period of about 8 h. The discharging process was interrupted for each tomogram and labelled with CR2-00 from the pristine to CR2-07, the fully discharged SoC.

Furthermore, due to the fast Li insertion, the outer electrode winding is moved axially outwards by the rising mechanical stress in the cathode, which leads to an obvious loss of Li conductivity. The strong mechanical degradation effects are suspected as a result of the high discharge rate of over −500 mA and a lower electrode compression in the top axial region, which supports an easy Li intercalation.

It should be noted that during the discharging process the excess electrolyte, which is mainly stored in the middle core of the cell, is consumed where a part may be used for forming an electrode electrolyte interface (EEI) on the Li electrode[35]. Furthermore, in the $LiMnO_2$ electrode, the electrolyte is soaked in the appeared cracks and voids, which makes the detection of those pores more difficult.

The excess electrolyte helps to compensate the electrolyte consumption and maintain the Li-ion conductivity between the electrodes during operation. During discharging, the excess electrolyte is consumed steadily and disappears from the inner region of the cell.

Near the axial top of the cell, the presence of electrolyte diminishes from the current collector wires after a discharge of about −225 mAh, whereas the electrolyte between the outer electrode winding and the casing diminishes slightly until ca. −750 mAh (see Supplementary Figs. 5 and 6). Furthermore, electrolyte is filling the in the first discharge period dried 2 mm region of the axial top electrode section after about −449.38 mAh. In general, after a discharge of −750 mAh, most of the excess electrolyte in the radial middle section of the cell seems to have moved into the cathode cracks or is filling the gaps left from the removed Li.

In the axial middle and bottom part, the Li removal from the anode can be seen by the thinning of the anode during discharge. The distinction between Li and the surrounding electrolyte is challenging at first, but with the sustained electrolyte consumption, it becomes more apparent as the discharge proceeds. In the pristine state, the Li metal foil is homogeneous and is very heterogeneously removed over the first part of the discharge process initiated by a quick Li intercalation in the upper axial section as mentioned above. In the further discharging process, the axial middle and bottom sections show an obviously thinner Li-electrolyte assembly on the opposite side of the current collector tab (see Supplementary Fig. 5) and almost all of the Li metal was consumed in the middle part of the discharged cell near the tab, whereas the Li distribution on the opposite, left and right side from the tab is still homogeneously distributed. However, due to the similar attenuation of the electrolyte and Li, regions may exist where Li is removed, but electrolyte is filling the gaps.

During cell operation, the Li from the anode is intercalated into the crystal structure of the manganese dioxide electrode, which is detectable by an intensity increase in the $Li_xMnO_2$ electrode in the neutron tomograms. However, the intensity increase is weak, and fluctuations are difficult to observe by eye, especially through the electrode thickness. Virtual unrolling of the electrode assembly and flattening of the electrodes makes it possible to plot the intensity changes along the length of the electrode assembly from core to case. This was therefore applied to the data and is discussed in the subsequent section.

Both methods show a very small deviation of the measured battery component sizes, which qualifies the data for a more advanced data analysis techniques, discussed in the next section. Supplementary Figures 7 and Fig. 8 show the calculation of the standard variation of the thickness change in the battery steel casing over the discharge time for the X-ray and neutron tomograms. Both imaging methods show a standard variation smaller than 1.5 µm.

**Virtual unrolling of tomography**. A comprehensive spatial analysis of the spiral-wound electrode assembly is challenging. Therefore, to facilitate spatial quantification along the length of the spiral electrode, virtual unrolling of the datasets was applied. To determine the electrode thickness change during the discharging process, an arc region out of the electrodes was selected, which covers about 360° of the battery circumference and four electrode windings, as shown in Fig. 1d for the unrolling of the neutron tomogram. Subsequently, the high-intensity nickel mesh was removed from the unrolled X-ray data by a thresholding process, before the volumes were averaged over the electrode's height to get a mean value of the cathode expansion and the intensity change.

Because of the similarity between the greyscale values of the $Li_xMnO_2$ and the current collector in the neutron data, it was not possible to remove the mesh by a threshold. Hence, the mesh appears as a constant value in the further processed neutron images.

Figure 3 shows cut-outs of the processed data for the X-ray and neutron images at different SoC. The images display the radial outer electrode windings in the upper region. The X-ray image of the cathode reveals a rapid volume expansion over the discharging period from the pristine state (CR2-000) to the partly discharged state to −604.90 mAh (CR2-096). The average cathode volume increased by about +27%, from 389 to 495 µm thickness, while the intensity decreases due to the Li insertion.

The neutron images show a similar volume expansion and electrode movement during discharging. The volume change for the discharged cell to −745.08 mAh is +30%, from 386 to 501 µm in thickness, and +26.5% to − 580.55 mAh, which corresponds to the +27% measured in the X-ray datasets by a slightly higher capacity. Both measurements indicate that the SoC has a major impact on the cathode volume expansion; this is caused by Li insertion in the $Li_xMnO_2$ structure and significant electrode cracking. Both discharge processes show a linear cathode volume expansion with an equal volume increase for similar c-rates, seen in Fig. 4a. Line profiles over the electrodes, shown in Fig. 3, display a decrease of intensity of the cathode for X-rays and an increase for neutrons. In both cases, the intensity changes heterogeneously over the cathode with a lower intensity 'shoulder' for the X-ray and a high-intensity shoulder for the neutron tomograms at the radial outward-facing $Li_xMnO_2$ electrode side. This indicates higher Li intercalation at the outer electrode side, which can be explained by the higher compression of the material at the inner side, caused by the electrode roll up during cell manufacturing. The lower compression at the outer side allows easier Li intercalation and less resistance for electrode expansion. Figure 4b shows the almost linear intensity reduction for the X-rays and increase for the neutrons of the cathode during cell discharge.

To determine the Li intercalation over the $Li_xMnO_2$ electrode length, the interval of the unrolling software was reduced to the single electrode thickness and expanded in length over the whole electrode, as shown in the horizontal cut through the X-ray tomogram in Fig. 1d. Figures 5 and 6 represent three of the unrolled cathodes, from the pristine to the partly discharged state, of the X-ray and neutron datasets, respectively. In the case of the X-ray images, the nickel mesh was removed via thresholding to avoid corruptions of the $Li_xMnO_2$ by the different intensity values.

In Fig. 5 the intensity of the X-ray data is averaged over the electrode height and for a better signal-to-noise ratio (SNR) every nine slices through the electrode thickness were combined, starting from the inner electrode side. The intensities of the electrode along the electrode length are then plotted in Fig. 5b for the pristine state (CR2-000) and partly discharged states (CR2-036 and CR2-096).

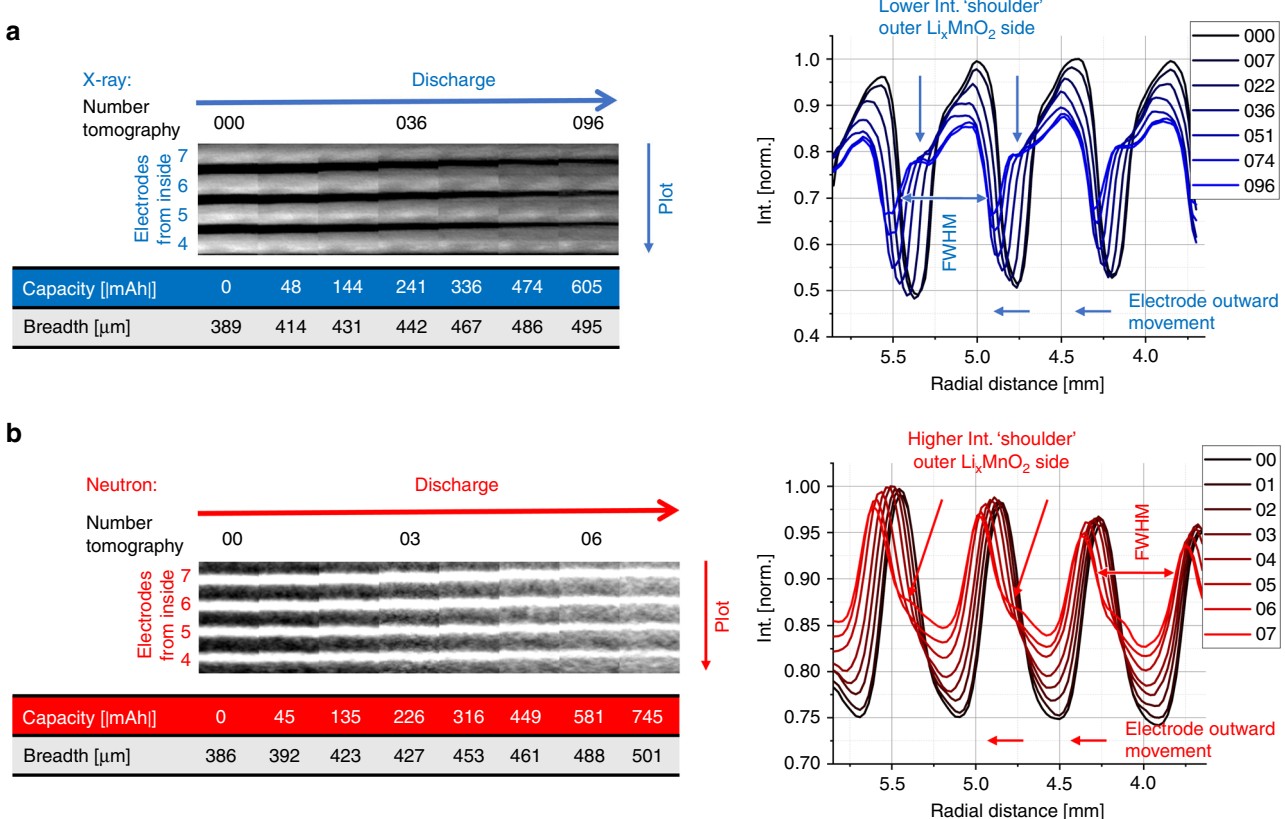

**Fig. 3 Cathode thickness and intensity change during discharging.** **a** shows cut-outs of virtual unrolled multilayer sections of the measured X-ray tomograms at different SoC from the pristine state (000) to the partly discharged state (096) to −604.90 mAh. The thickness increases of the $Li_xMnO_2$ electrode, during the discharge, is about 27% measured by the full-width at half-maximum (FWHM). The intensity plot shows a higher lithium intercalation in the cell outward electrode side by a lower intensity 'shoulder'. **b** shows a similar behaviour for the discharged cell imaged by neutrons. The electrode swelling is about 30% for a cell discharge to −745.08 mAh (07) and 26.5% to 580.55 mAh (06), which shows a similar electrode expansion as for the X-ray tomograms due to the similar discharge conditions. The plots illustrate the complementarity between X-rays and neutrons.

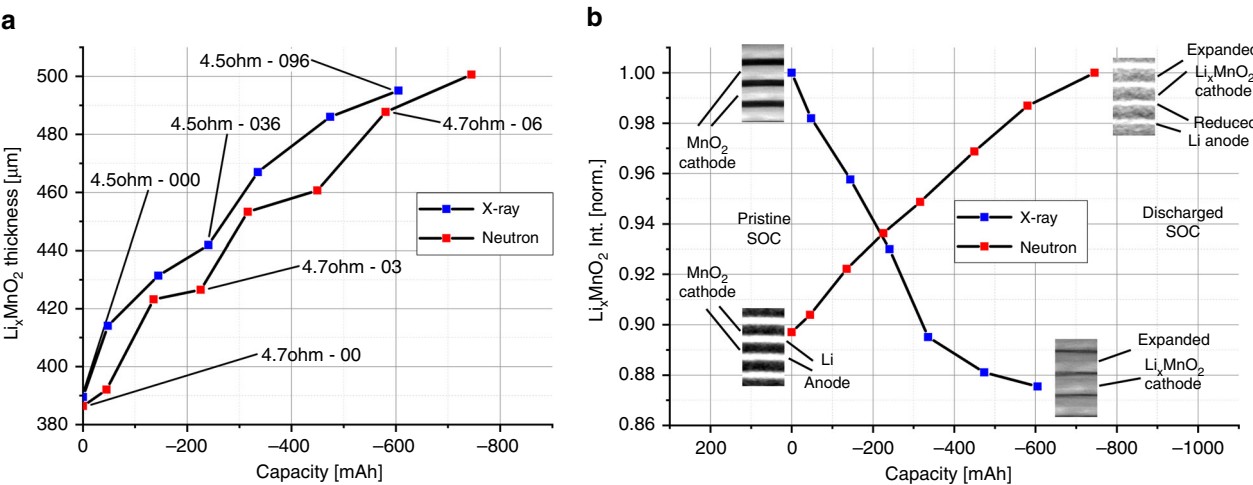

**Fig. 4 Cathode thickness and intensity variation vs. capacity.** Plot **a** shows the $Li_xMnO_2$ electrode thickness increase by the lithium intercalation during cell discharging by similar c-rates. For similar c-rates, the electrode thickness increases uniformly. But it seems that a slightly higher c-rate (blue) causes a higher increase in the volume expansion in comparison to lower c-rates (red). The graphs in **b** compare the intensity change of the $Li_xMnO_2$ electrode during discharging. The lithium intercalation process in the electrode causes an intensity decrease for X-rays and an increase for neutrons (for comparisons see Fig. 3).

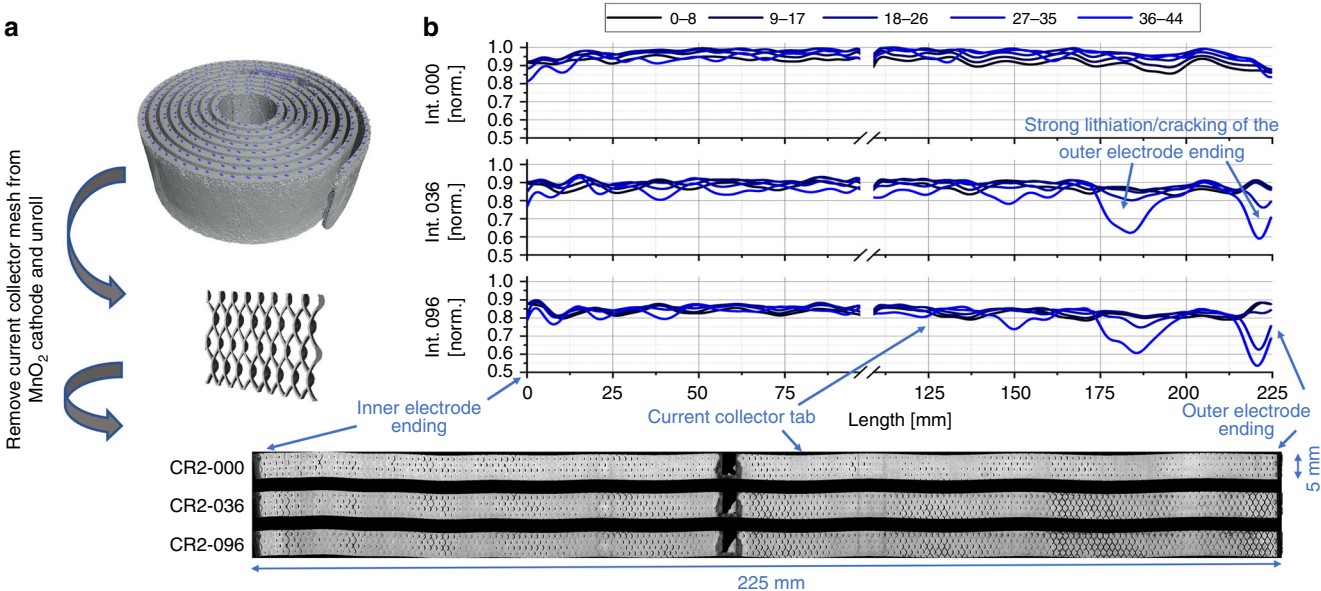

**Fig. 5 Lithium distribution at various depths of the Li$_x$MnO$_2$ electrode during discharging, using unrolled X-ray tomography data of the upper part of a CR2 cell.** For the analysis of the lithium distribution over the MnO$_2$ electrode thickness, **a** the current metal collector mesh is digitally removed before the electrode is unrolled. The electrode thickness is divided into five depths and the normalised grey values plotted over the electrode lengths. **b** represents the changed lithium distribution for the tomograms CR2-000, CR2-036 and CR2-096. The lithium content or cracking increases much more on the outer electrode side (blue) compared to the inner side and the electrode outer ending, as indicated by the reduced grey values. The lithium content is highest at the side of the current collector tab, exhibiting sinusoidal variation with increasing amplitude as the cell discharges.

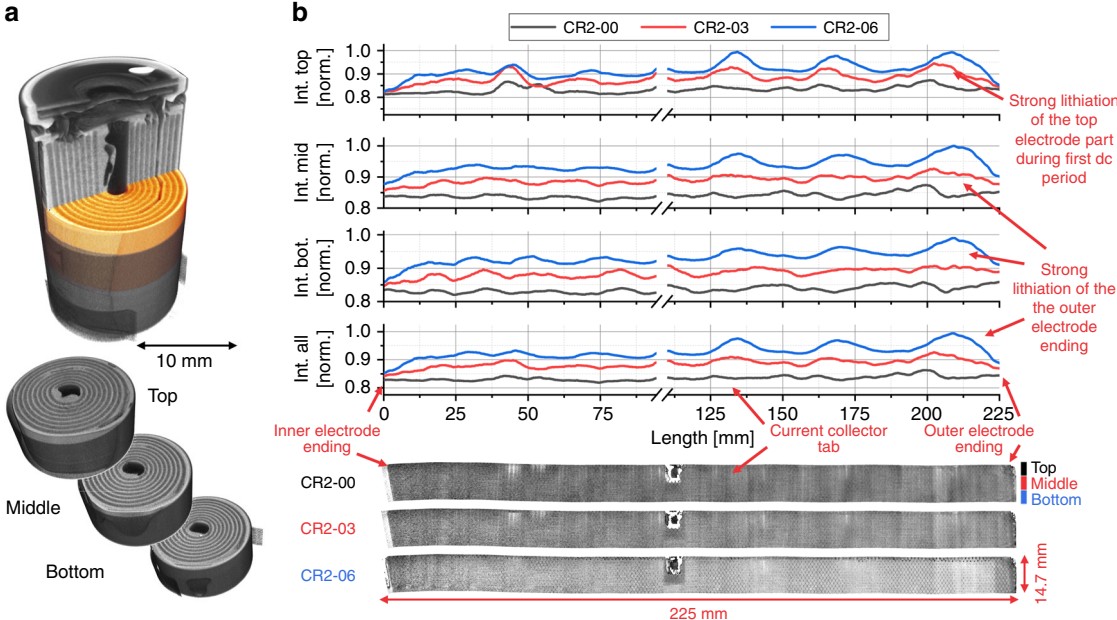

**Fig. 6 Organisation of electrode unrolling (top, middle, bottom) and analysis of the lithium distribution in the Li$_x$MnO$_2$ electrode during discharging using neutron tomography data.** The analysis of the lithium intercalation in the Li$_x$MnO$_2$ electrode is divided in four sections, the top, middle, bottom and overall part. **a** shows the pristine neutron tomogram divided in the studied parts. **b** displays the unrolled and over the thickness averaged electrodes of the pristine (CR2-00), partly discharged to −225.71 mAh (CR2-03) and partly discharged to −580.55 mAh (CR2-06) SoC and the related line plots. The intensity increase, due to the lithium intercalation, during discharging is clearly visible. On the top, the electrode undergoes a very strong lithiation during the first period of discharging. Over the electrode length, a sinusoidal intensity profile with increasing period is observed, with maxima at the tab side. This structure is caused by a higher lithium content on that side.

The overall intensity decreases during the cell discharge due to the Li intercalation in the MnO$_2$ structure and the cracking of the electrode. A higher intensity change is clearly visible at the outer windings of the electrode (left side from the current collector), with a decrease observed over the electrode length inwards, which indicates higher Li intercalation or cracking caused by a lower compression of the active material. Furthermore, at the radial outer cathode layer an inhomogeneous intercalation process is detected, in the form of a sinusoidally shaped intensity variation with increasing amplitude to the outer electrode ending. The

number and positions of the local intensity minima are consistent with the number of the electrode windings and are positioned at the left side from the current collector tab radial inward looking, see Fig. 5b. It can be concluded that at this side, the compression of the active material is less, and favoured an easier intercalation/cracking followed by a simultaneous volume expansion. Compression may vary due to the current collecting tab anchoring the electrode assembly against the steel casing on that side and benefits the porosity increase at that side during operation. The larger intensity reduction at the outer electrode side can be explained by being less confined and more susceptible to expansion.

After unrolling the neutron datasets, the electrodes were split into three sections at different heights of about 4.8 mm, each labelled with top, middle, or bottom, see Fig. 6. The slices were then combined through the electrode thickness to obtain a higher SNR. The same procedure was carried out for the whole electrode to analyse the intensity variation at the three different electrode heights of the cell and compare them to the mean value taken from the bulk.

Figure 6b shows the averaged electrodes of the pristine (CR2-00), partly discharged to −225.71 mAh (CR2-03) and the more discharged to −580.55 mAh (CR2-06) SoC with the corresponding greyscale intensity line plots. The highest average intensity increase of the three sections is detected at the cell top section for the first period of the discharging process (CR2-00 to CR2-03), which seems to be the result of the less compressed electrodes, which facilitate a better Li-ion intercalation in the axial upper 2 mm high cathode section as discussed in the neutron CT section above. In the further discharging process, the middle and bottom part of the battery cell show an increase in Li insertion, whereas the upper part shows a stagnation in Li intercalation due to the high degree of Li removal from the anode and the missing electrolyte from the first period of the discharging. After discharging to −580.55 mAh (CR2-06) all three sections display a similar amount of intensity increase, which means that the two axial lower sections compensated the high Li insertion of the top section from the first discharge period in the second. In corroboration to the X-ray images, the neutron data shows the sinusoidal intensity to be most prominent in the more discharged states with the highest intensity/Li increase at the current collector tab side. This indicates that the position of the minima and maxima of the sinusoidal intensity variation over the electrode length vary from cell to cell, and may depend on the local electrode compression from the manufacturing process. The local intensity increase is also strong related to a local volume increase of the cathode.

An irregular behaviour of the Li intercalation at the inner and outer electrode windings is observed in Fig. 7. During the cell discharge the X-ray images show no Li intercalation in a range of about 5 mm for the inner cell winding and in a range of about 6 mm for the outer cell winding, see Fig. 7a. A possible reason is that the Li metal electrode stops before reaching the inner and outer endings of the wound $Li_xMnO_2$ layer, as previously reported[25]; the Li metal electrode is shorter in length than the $Li_xMnO_2$ electrode.

The $Li_xMnO_2$ electrode went beyond the Li electrode resulting in underutilized $MnO_2$ at the ends. The same behaviour is identified for about 5 mm of the inner and 7 mm of the outer electrode endings in the discharged neutron tomograms, see Fig. 7b. But here, the Li intercalation is prevented at the outer electrode ending until the last two discharge steps from about −580 to −745 mAh. Consequently, the most likely mechanisms for the Li insertion in the electrode endings is Li diffusion through the $Li_xMnO_2$ electrode or that a longer transport through the remaining electrolyte becomes more likely at a lower SoC assumed the endings are sufficient wetted by electrolyte.

However, despite the behaviour of the two cells investigated here being very similar, we cannot say for certain that their behaviour is statistically significant and representative of all cells operated under similar conditions.

## Discussion

4D CT data were obtained from two primary CR2 Li/MnO$_2$ cells while they were discharged under identical conditions. Due to the complementary nature of the two probing methods, one cell was examined using operando X-ray CT and the other using in situ neutron CT. The multi-modal characterisation was then assessed using a novel virtual electrode unrolling technique.

X-ray CT allowed for the quantification of mechanical degradation effects such as electrode cracking from the electrode bending process during cell manufacturing. Whereas neutron CT gives information about the electrochemistry such as Li-ion transport and intercalation, electrolyte wetting of the cell and consumption or gas formation by, for example, electrolyte decay.

Due to the limited interaction with light elements, Li insertion could be detected only indirectly by X-rays, through an intensity decrease in the $Li_xMnO_2$ electrode. Neutron imaging enables the direct detection of the Li diffusion from the Li metal electrode to the $Li_xMnO_2$ electrode. Consumption of excess electrolyte is observed and is thought to be due to electrolyte reaction to form the EEI layer, as well as filling the emerging cracks in the cathode as it expands upon lithiation. More pronounced electrode cracking and delamination, as a result of lower electrode compression, was detected at the top section of the electrode assembly, as well as the outer electrode windings. The neutron data showed a very high activity in the top 2 mm of upper axial electrode section in the first period of the discharge, which resulted in a high Li intercalation and electrolyte consumption, up to the complete consumption of the Li from the anode and removal of the electrolyte. The electrolyte-dry areas are wetted in the second period of the discharging where that regions show a lower electrochemical activity.

To identify weak fluctuations in intensity caused by Li insertion, electrode wetting or mechanical degradation, virtual electrode unrolling was performed on multiple electrode sections and over the full length of single electrodes. Multiple sections of the electrode were unrolled and used to determine the linear average electrode thickness change. It is suspected that the electrode thickness increased with growing c-rate as a result of stronger electrode cracking. A more favoured Li intercalation was detected at the outer $Li_xMnO_2$ electrode side due to a weaker compression, as a result of the electrode bending of the active material. Additionally, a heterogeneous sinusoidal-shaped intensity variation over the $Li_xMnO_2$ electrode length was found, with a favoured increase in Li insertion at the outer cell windings caused by lower compression of active material and a higher Li conductivity by better electrolyte wetting. Finally, lithiation was not detected at the inner and outer cathode endings, which was caused by the Li metal electrode ending before reaching the cathode end. Due to the electrolyte and Li having similar attenuation coefficients for neutrons, a direct correlation between the SOC and the amount of Li consumed could not be made, highlighting a limitation of this technique for the most commercially relevant cells.

The combination of X-ray and neutron imaging modalities provides clear benefits for the interpretation of active electrochemical materials, which contain many components with different X-ray and neutron attenuation properties. We expect that this methodology will be valuable for studying and optimising a range of battery chemistries, not limited to the primary cell studied here; it will also make great benefits in the field of rechargeable cells. Moreover, the unrolling protocols

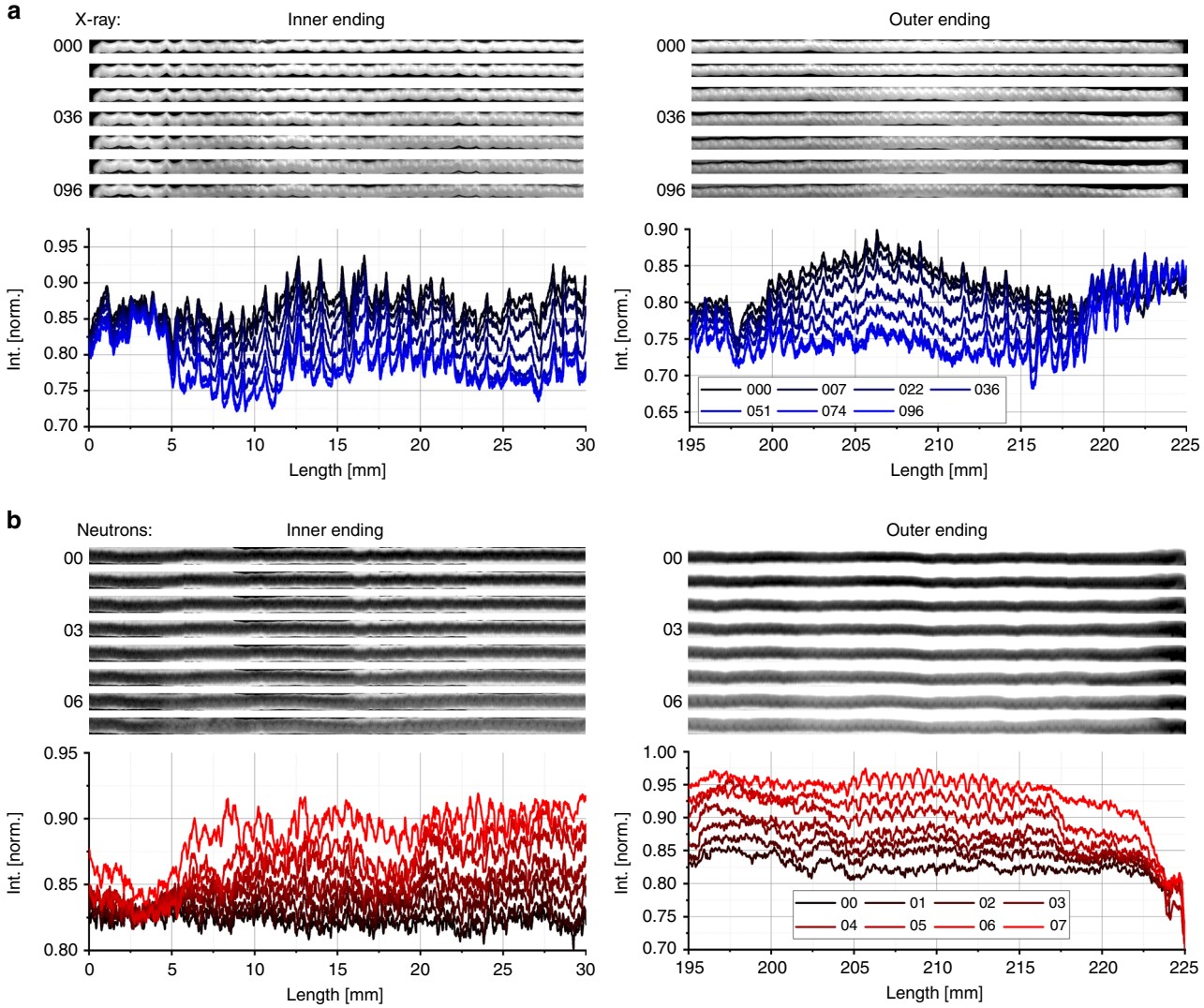

**Fig. 7 Electrode endings and middle part at different SoC.** The images and line plots in **a**, **b** show the attenuation of the $Li_xMnO_2$ electrode during cell discharging for the X-ray and neutron imaged cells, respectively. In both cases, the inner electrode parts show an almost homogeneous lithiation (not displayed here). The electrode endings do not show any lithiation activity for the X-ray scans and late start of lithiation in case of the outer electrode endings for the neutron scans.

demonstrated here provide a new means for direct correlation and comparison of multi-modal image data, which aid the straightforward interpretation of spatial trends that are otherwise challenging in a spiral-wound configuration. Furthermore, these data present an example of the powerful insight that may be gained from 4D imaging; the bulk properties that computational models were previously based on may now be replaced by spatially resolved, transient values. The implications of this lie in that models would not need to apply the macroscopic assumptions made by averaging bulk properties, and consequently, due to the capability of incorporating local heterogeneities, they would be able to provide a more accurate and comprehensive description of the practical operation, degradation and failure of a cell.

## Methods

**Experimental.** For the study of the $Li^+$ separation from the Li metal anode and the intercalation in the $MnO_2$ cathode of a commercial CR2 primary cell X-ray and neutron attenuation CT were used, where the imaging contrast is a contribution of the absorption and scattering cross-sections of the elements. In both experiments, the CR2 cells were discharged over a resistor of 4.7 Ω. However, the resistivity used for the X-ray measurements was at a value 4.5 Ω due to the fluctuations given by the manufacturer.

High-speed X-ray tomograms were collected at the ID15A imaging beamline at the ESRF. For the X-ray experiment, the battery was discharged over a 4.5 Ω resistor and the upper cell part was imaged during 2.8 s of total exposure time every 40 s with a total number of tomograms of 103. The tomographies are labelled from CR2-000 to CR2-102. For the current investigation, seven tomograms were used for the data analysis starting with the pristine SoC at the discharge time 0 s and expanding time intervals between each tomography (300 s, 900 s, 1500 s, 2100 s, 3000 s, 3900 s). The tomograms are marked by blue squares in the discharge curve in Fig. 1b. Figure 2a shows orthogonal slices of three of the 3D reconstructed X-ray tomograms, in the pristine state, partly discharged to −240.65 and −604.90 mAh.

Neutron tomograms were taken at the V7 CONRAD-2 imaging beamline at HZB. Tomograms were collected in the pristine SoC and after with an increasing time interval between each tomography (300 s, 900 s, 1500 s, 2100 s, 3000 s, 3900 s, 5100 s). The discharge process was interrupted during tomography acquisition due to the long exposure time of 8 h per tomogram. Based on the larger field-of-view (FOV), the whole cell size could be imaged which is displayed in Fig. 2b, as orthogonal slices of the reconstructed 3D images for different SoC. The discharge profile is given in Fig. 1c and shows all eight discharge steps, here labelled with CR2-00, for the pristine state, to CR2-07 for the almost fully discharged cell: red squares mark the SoCs where neutron tomograms were recorded.

3D images can give a comprehensive insight into the structure, working principle and morphological changes of a cell. However, it can be challenging to extract meaningful spatial information from a spiral-wound structure. To facilitate interpretation of the spatial data along the electrode length, a plug-in from the Amira Software Package was applied, which was created by Konrad-Zuse-Zentrum Berlin (ZIB)[34] for virtually unrolling the spiral-wound structure and take

measurements along its length. This enables analysis of morphological changes and degradation processes inside the battery electrodes as well as on its surface, and more direct correlation and comparison of the multi-modal image data. An example of an unrolled section is shown in Fig. 1d and a detailed description of the technique is given below.

**CR2: Li/manganese dioxide**. The commercial CR2 primary battery from Duracell[26] is a spirally wound cylindrical cell, made for high-pulse and middle-rate applications, with an electrode layer sequence of '…$Li_xMnO_2$|separator|Li|separator|$Li_xMnO_2$|separator|Li…' (Fig. 1a), where the thick porous $MnO_2$ cathode is pasted onto a supporting nickel current collector mesh. The Li metal anode functions as its own current collector, and both electrodes are separated by a microporous polypropylene Li-ion-permeable separator. All cell components are packed in a steel container and are soaked with a non-aqueous, Li-ion-conducting, organic electrolyte. During the discharge process, Li is oxidised at the Li metal anode, which involves the loss of electrons and the release of $Li^+$ ions into the electrolyte. The $Li^+$ ions pass through the Li-ion-permeable, electrolyte soaked separator and intercalate into the crystal structure of the porous $Li_xMnO_2$ cathode by the reduction of $Li^+$ with the freed electrons from the anode. The Li intercalation in $MnO_2$ is a multiple discharge process with different stages of $MnO_2$ reduction described by Nardi et al.[36] and involves a phase transformation from the tetravalent $Mn^{IV}O_2$ to the trivalent $Mn^{III}O_2$ with a noticeable increase in volume. The overall reaction is given by:

$$x\text{Li} + \text{Mn}^{IV}\text{O}_2 \rightarrow \text{Li}_x\text{Mn}^{III}\text{O}_2. \tag{1}$$

**High-speed X-ray CT**. The X-ray imaging experiments were carried out with a 76 keV monochromatic X-ray beam on the high-speed X-ray imaging beamline ID15A at ESRF[6,27]. This beamline has a high photon flux and brilliance, which enables imaging with a high spatial and temporal resolution. The used camera system included a PCO Dimax (PC AG, Germany) high-speed camera, which allowed recording of 12-bit images with a pixel size of 10.87 μm and a pixel matrix of 794 × 800 pixel, which limited the FOV to 8.6 × 8.7 mm$^2$ (horizontal × vertical). Due to the restricted FOV, only a half of the battery was imaged with the centre in the middle of the cell. To capture the cell fully in the lateral direction 'extended FOV' tomography was used. For that the centre of rotation was placed close to the edge of the FOV and 360° tomograms composed of two 180 half tomograms with 2 × 2000 half projections and an angular increment of 0.09° were taken. One tomogram was made every 40 s with a total acquisition time of 2.8 s and 0.7 ms per radiograph. During the experiment, the CR2 battery was discharged with a 4.5 Ω resistor (see Fig. 1b). The electrical contact between the constant resistor circuit and the battery was made by using an electrical slip ring (P4 + Compact Slip Ring, Moog, UK) built into the ID15A rotation stage (ABR1000, Aerotech, USA), which allowed continuous rotation. The electrical connection was made by a Cu foil pressed against the positive and negative battery terminals and tightly wrapped with electrical tape. Electrical wires were soldered to the Cu tabs and connected to the slip ring and the resistor. To minimise beam hardening artefacts, a finely woven X-ray transparent Cu mesh was connected to the positive terminal tab, at the battery head, and fixed sideward, through the FOV, along the battery casing down to the cell bottom. After battery discharging and image recording, all 103 recorded tomograms were reconstructed by using standard reconstruction software and filtered back projection (FBP) reconstruction algorithm. A more detailed description of the measurements can be found in the authors previous report[25].

**High-resolution neutron CT**. The tomographic neutron imaging was carried out on the V7 CONRAD-2 neutron imaging beamline at the BER II research reactor at HZB[28,29], using a polychromatic cold neutron beam collimated by a pinhole with an L/D from 500 and a neutron flux of $2.7 \times 10^6$ n cm$^{-2}$s with a wavelength maximum at 2.58 Å at the imaging position. Radiographic projections were recorded by an indirect neutron imaging detector with a 10 μm GADOX (gadoliniumoxisulfite, Proxivision, Germany) scintillator screen and CCD camera (DW436N-BC, Oxford Instruments Andor, UK) in combination with a bright infinity-corrected optics using TV-Heliflex lens (Rodenstock f = 50 mm, f/1.07, Numerical Aperture = 0.42) and Nikon photo lens (Nikkor 50 mm 1:1.2, Nikon, Japan) as described in[37], which was employed at the middle measuring position of the instrument where beam conditions are optimised for neutron flux and high spatially resolved imaging. The CR2 Li-ion cell was positioned in the front of the scintillator screen and filled the complete 26 mm diameter FOV, which was given by the rim of the photo lens. The camera array has 2048 × 2048 pixel with a pixel size of 12.9 μm. Due to the highly neutron-attenuating hydrogen content in the outer plastic shell, the CR2 battery shell was removed for a better neutron transmission before it was mounted on the V7 rotation stage (Goniometer 409, HUBER Diffraktionstechnik, Germany). The cell was mechanically connected by a cylindrical TEFLON holder, around the lower battery casing, which was screwed to an insulating PTFE cylinder fixed on the rotation stage. The negative electrical contact was made by a copper wire glued with aluminium tape to the battery casing, which contacted the negative terminal, and by a cable soldered to a nickel-plated crocodile clip connected to the positive terminal at the cell head. Both cables were led away downwards and upwards, respectively, out of the FOV and connected to a potentiostat (1000E, Gamry Instruments, USA). The first tomogram was taken in the pristine state before discharging and the following datasets were collected consecutively with an increasing time interval between each tomogram. The discharging process was made over a 4.7 Ω resistance and interrupted for about 8 h for taking tomograms (see Fig. 1c) over a 360° rotation with an angular increment of 0.45° and 800 steps. For each angular step, three radiographies were made with 9 s of exposure time and 5 s readout time of the camera. The three radiographies where combined to one projection by a median filter to reduce white spots, caused by scattered or fast neutrons and secondary γ-rays, which hit the camera chip. The data collection and discharging processes were controlled using trigger signals between the beamline control software (LabVIEW, National Instruments, USA) and the potentiostat. For data processing, the 'Remove Outliers' and median filter of ImageJ Fiji[38] were used to clean the 16-bit images from remaining white spots 3D reconstructions were performed using Octopus 8.9 software (XRE, Gent, Belgium) with FBP reconstruction algorithm and beam hardening correction.

**Virtual unrolling of the battery electrodes**. The access to uneven, folded or rolled surfaces within a sample is hardly possible. It is generally possible to extract small flattened areas out of a 3D volume, but these areas mostly do not deliver meaningful results. To overcome this, we applied a plug-in for Amira (Amira 6.5, ZIB/Thermo Fisher Scientific, Germany/USA), which was created by the ZIB[34], to get access to our data. Originally developed for papyri unfolding project, we used this plug-in for battery unrolling. The working principle of unrolling this compound is shown in Fig. 1d and Supplementary Fig. 1. The electrode-separator compound within a battery of type CR2 is rolled. Consequently, each layer of the 3D volume represents a spiral-like course of this compound, see Supplementary Fig. 1b. A reconstructed 3D volume is given in Supplementary Fig. 1a. The experimentalist has to define certain contours in several slices of the 3D volume. These contours are depicted by the red line in Supplementary Fig. 1a. The same contours are linearly interpolated for slices in between, while the contours for axial outer slices are created by constant extrapolation. Therewith, the whole 3D volume is included in this reshaping of the data set. Taking all contours into account, a surface is spanned in the 3D volume. For example, a contour is given in Supplementary Fig. 1b as a red line. If desired, the line can be spread to an interval, as marked by the white lines. For each point on the red line, perpendicular line segments are created by the plug-in, and consequently, neighbouring surfaces are created. All points along these line segments of the 3D volume are subsequently written into a new volume data set, which contains the unrolled volume of the battery. A cross-section of the unrolled volume is given in Supplementary Fig. 1c. An average of several slices usually reveals a good signal to noise Supplementary Fig. 1d. A reslice of the unrolled volume given in Supplementary Fig. 1e, showing a single component of the battery in a flat shape such as the nickel current collector mesh. The latter step was done using ImageJ[39].

## Data availability
The data supporting this study can be made available from the corresponding author upon request.

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

## Acknowledgements

This work was carried out with funding from the Faraday Institution (faraday.ac.uk; EP/S003053/1), grant number FIRG001. We would like to acknowledge the Royal Academy of Engineering (CiET1718\59) for financial support. This work was authored in part by the National Renewable Energy Laboratory, operated by Alliance for Sustainable Energy, LLC, for the U.S. Department of Energy (DOE) under Contract No. DE-AC36-08GO28308.

## Author contributions

R.F.Z. and P.R.S. conceived the study. R.F.Z. led the experimental work and analysis. T.A. and D.B. contributed the virtual electrode unrolling of the neutron and X-ray tomography data. D.P.F. provided the 4D X-ray CR2 datasets, measured on the ID15 beamline at ESRF, and helped with the data analysis. N.K. and H.M., beamline scientists on the CONRAD neutron imaging beamline at HZB, supported the neutron imaging of the CR2 cell and did the 3D reconstruction of the measured data. A.T., beamline scientist on the D50 neutron imaging beamline at ILL, helped to perform the 4D scans of the CR2 cell. T.M.M.H., I.M., W.K., D.J. L.B., and P.R.S. supported the work with their expertise in battery science and X-ray, and neutron imaging, and helped to design the experiment and analysis. All authors have contributed to writing and reviewing the manuscript.

## Competing interests

The authors declare no competing interests.
