## [Peer Review File · Nature Communications]

Reviewers' comments:

Reviewer #1 (Remarks to the Author):

4D imaging of Li-batteries using operando neutron and X-ray computed tomography in combination with a virtual unrolling technique

This is an extremely interesting contribution to battery science made possible by the combination of two imaging techniques applied to a battery undergoing controlled discharge. The work shows procedures for inspecting electrode consumption spatially, hence will provide a route to more efficient battery construction. The "virtual unrolling" developed herein is a useful data analysis tool. Minor suggestions are made below with respect to achieving a consistent coordinate system.

Notes:

line 13: "high-resolved" should be "high-resolution"

line 54: "known for its innovation in neutron imaging" This is an opinion. Please replace with a factual statement such as "known for its combination of high-resolution neutron imaging with step-wise sample evolution "

line 63: Fig 1d. The chirality (handedness) of the X-ray and neutron images differ. This is a known problem with state-of-the-art imaging systems where detectors are frequently exchanged. One of the data sets should be reflected about an axis. It would be helpful to rotate a data set about the axial axis (see triad in Fig 1d) to mutually register the spiral electrodes in the X-ray and neutron images.

line 63: Fig 1a. The oblong objects in the cathode denote the "nickel current collecting mesh" (line 83). If a short note can be added to the already long Fig 1 caption to identify the oblong objects, that would be helpful.

line 145 "A higher rate of lithium consumption ..." This is a very interesting observation. Do the authors know the stoichiometric excess of lithium in this battery? That is, for the completely discharged battery, will 50% of initial lithium remain as unreacted lithium metal? An eyeball integration of the neutron images suggests 50% unreacted lithium.

line 180 "The neutron images show a smaller volume expansion and electrode movement during discharging." This brings up the point of sample statistics. The paper does not clearly state that two different batteries are imaged. There is no attempt to show that batteries are equivalent based on mass or electrical capacity.

line 194: Fig 3b line plot. Tomogram 07 is not in the stacked line plot. Just curious, did the switch from cold to thermal neutrons create too much offset for normalization to the same scale as the other line plots?

line 194: Fig 3. The x-axis in the line plots is (I think) a radial distance. Would it make more sense to set the origin of this distance to the battery axial center? We note that Fig 5 uses "length" as distance along an electrode.

line 219: Let's consider -045 image and line plots. The dips in the line plot at slices 36-44 at about 160 mm and 190 mm should be visible in the -045 image. I think I can see the changes, but a call out with color rectangles (as done in Fig 2b "regions of strong...") would be helpful.

line 228: "result of the better Li-ion conductivity caused by the excess electrolyte at the bottom". This

points out the author's failure to clearly establish a coordinate origin for the electrode length. I thought origin was at the center per the axis triad in Fig 1d. Now, it appears the origin is the outer layer. This is confusing.

line 250. The point of Fig 7, activity of electrodes at the very center of the battery, is obscured by the confusion of whether center of battery is a 0 mm or 220 mm. Interesting, the comment "late start" is a powerful indication of the complementary use of X-ray and neutron imaging.

Great work folks!

Les Butler
Louisiana State University
USA

Reviewer #2 (Remarks to the Author):

The manuscript by Ziesche and co-workers reports the temporally and spatially resolved imaging of a primary battery undergoing discharge using a combination of x-ray and neutron tomography. The sensitivities of x-ray to heavier elements and neutron to lighter elements complement each other in measurements, yielding a clearer picture of the structural evolution of various components in a primary battery with rolled electrodes. This study is a nice extension of the previous effort in applying in situ x-ray and neutron measurements to studying battery cells in operando, in particular the virtual unrolling representation of layered structures allows for semi-quantitative analysis of structural variations across the cell and over the time.

As this Communication reports extending the application of a novel measurement methodology developed in the past years to studying batteries in operando, several noticeable deficiencies need to be addressed to elevate the caliber of the study using advanced techniques in major research facilities.

1. The reported measurements are not real-time in-operando for neutron, and not truly multimodal or simultaneously probing the same location and the same event of the structure evolution toward achieving the goal of synergistic measurements of hard/soft matter. To take advantage of the complementary compositional sensitivity of x-ray and neutron, it is desirable to apply measurements to a single cell, or at least identical cells operating at identical conditions.

2. Thus the comparison and complementarity are compromised with the high c-rate in x-ray measurements and low c-rate in neutron measurements. There are unique limitations and advantages of each technique, overlap of certain operation conditions are needed.

3. The outlier data point using thermal neutron could be addressed with more rigor, and better quantification is possible through normalization using the flux and cross-section ratios of thermal and

cold neutrons.

4. The statistical significance of the data and observation is not sufficiently addressed. The extents of the generalization of the failure mechanism as observed in this study are not assessed.

5. It is necessary for better quantification of experimental measurements to be more relevant to the electrochemical characteristics of batteries. The comparison between the amount of transported Li and the state of the charge and the correlation between the volume change and the change of Li content could be addressed.

6. In general, the focus of the study is mostly on engineering aspects of primary batteries. It is desirable to apply novel measurement techniques to the state of the art battery materials or new battery architectures and addressing the most relevant scientific issues in advancing battery technology.

Recommendation -- the manuscript could be further improved. The scheme and outcome could fit to a journal more specialized on methodology or toward electrochemical research community.

Reviewer #3 (Remarks to the Author):

This is an interesting paper detailing a novel combination of imaging methods to extract insightful information about the battery discharge process. This kind of work is an important contribution towards the goal of understanding the connections between manufacturing/production methods and discharge performance. The method gives significant insight into transient, spatially resolved behaviors relevant to battery performance. While not done here, using these techniques to study transient discharge behavior in different battery designs could offer impactful new insights as to why some designs or manufacturing methods result in superior capacity/cycle life, which in many cases can currently only be hypothesized with the help of end-state battery imaging or computational modeling. I suggest some minor revisions before publication:

1. Lines 178-182:

a. Can you be sure that the differences in electrode expansion between medium and high rate discharges are due to cracking and not due to differences in imaging technique and/or image postprocessing? Is resolution sufficient enough to confidently report these volume expansion numbers without more statistics? Even a +- reporting of spread throughout the length of the electrode to accompany the average value reported would give these measurements more weight. It may be stated elsewhere, but how does the pixel size compare to the volume expansion: how much error is introduced if one pixel of expansion is incorrectly included/not included in the expansion measurement?

b. What is the theoretical/measured expansion of pure LiMnO₂ upon lithiation and how does that compare to these measured values? This may give insight into how much porosity (and therefore Li+ transport through the pore network) is decreasing due to swelling, at least in the medium C-rate case where there is less cracking.

c. You chose to report averaged volume increase, yet for the other metrics in subsequent figures you show the values as a function of length around the spiral cathode. Can you state why? Is the volume increase uniform around the cell or was there sinusoidal behavior like you see in Figure 6? For example, do you see more significant electrode expansion opposite the tab current collector to support the conclusion in line 214/215 and 218?

2. Line 190-191: Since this statement is repeated several times throughout the paper during the analysis, it should be more detailed by explaining why this statement is true (A lower compression results in a more porous electrode, allowing for easier lithium diffusion through the pore network as well as providing more room for mesoscale electrode expansion into the pores as well as macroscale expansion via cracking of a less dense/strong composite material.) Up to this point in the paper, there has been no mention of electrode porosity, so it may not be clear why the statement is true.

3. Line 249. You mention mechanisms, but only state one mechanism. Lithium may not necessarily have to diffuse through the cathode material to get to the cathode endings. Early in discharge, there may be no incentive for the lithium to transport out to the cathode endings through the electrolyte given that there is much closer cathode material available. As the state of charge of the nearby cathode material increases and approaches capacity, the longer transport through the electrolyte to the cathode endings (low SOC) likely becomes the most favorable path. I would think a longer transport path through electrolyte to low-SOC cathode is the likely mechanism here, rather than lithium diffusion through the active material as you state.

4. While not directly related to this work, the impact on computational battery modeling could be mentioned somewhere in the introduction/conclusions. The transient, 3D spatially-resolved image data of lithiation and electrode swelling presented here would be invaluable to the modeling field that is quite starved for such detailed model validation data. For transient discharge simulations, typically only global measurements are available (discharge curves, macroscale pouch swelling, etc.).

Response to Reviewers' comments:

Reviewer #1 (Remarks to the Author):

4D imaging of Li-batteries using operando neutron and X-ray computed tomography in combination with a virtual unrolling technique

This is an extremely interesting contribution to battery science made possible by the combination of two imaging techniques applied to a battery undergoing controlled discharge. The work shows procedures for inspecting electrode consumption spatially, hence will provide a route to more efficient battery construction. The “virtual unrolling” developed herein is a useful data analysis tool. Minor suggestions are made below with respect to achieving a consistent coordinate system.

We greatly appreciate the reviewer’s time on this manuscript and extend our thanks for the constructive comments below.

Notes:

line 13: “high-resolved” should be “high-resolution”

We have changed ‘high-resolved’ to ‘high-resolution’, see line 16:

“High-speed X-ray tomography; high-resolution neutron tomography; virtual unrolling; lithium distribution; lithium transport; electrode degradation”

line 54: “known for its innovation in neutron imaging” This is an opinion. Please replace with a factual statement such as “known for its combination of high-resolution neutron imaging with step-wise sample evolution”

Thank you for your comment, we replaced ‘known for its innovation in neutron imaging’ with your suggested term “known for its combination of high-resolution neutron imaging with step-wise sample evolution”, see line 56:

“... research reactor at Helmholtz Centre Berlin, known for its combination of high-resolution neutron imaging with step-wise sample evolution.”

line 63: Fig 1d. The chirality (handedness) of the X-ray and neutron images differ. This is a known problem with state-of-the-art imaging systems where detectors are frequently exchanged. One of the data sets should be reflected about an axis. It would be helpful to rotate a data set about the axial axis (see triad in Fig 1d) to mutually register the spiral electrodes in the X-ray and neutron images.

We mutually registered the X-ray and neutron images by their chirality in Fig. 1d and Fig. 2.

line 63: Fig 1a. The oblong objects in the cathode denote the “nickel current collecting mesh” (line 83). If a short note can be added to the already long Fig 1 caption to identify the oblong objects, that would be helpful.

In the caption of Fig. 1 we added an explanatory sentence which explains the ‘current collector mesh’ in the image:

“Clearly visible in the X-ray images is the nickel current collecting mesh which appears brighter than the Li_xMnO_2 active electrode material.”

line 145 “A higher rate of lithium consumption ...” This is a very interesting observation. Do the authors know the stoichiometric excess of lithium in this battery? That is, for the completely discharged battery, will 50% of initial lithium remain as unreacted lithium metal? An eyeball integration of the neutron images suggests 50% unreacted lithium.

We thank the reviewer for their comment. Due to the difficulty in distinguishing between the remaining electrolyte and lithium (which have a similar attenuation) in a fully discharged battery we are not able to give a quantitative value on the unconsumed amount of lithium. In future work, disassembling the discharged CR2 may provide some insight on this but since the cells used in this study have degraded beyond the final imaging experiment, it is not possible for this work. We appreciate the reviewers comment and hope that they understand challenge faced in providing a quantitative answer.

line 180 “The neutron images show a smaller volume expansion and electrode movement during discharging.” This brings up the point of sample statistics. The paper does not clearly state that two different batteries are imaged. There is no attempt to show that batteries are equivalent based on mass or electrical capacity.

We thank the reviewer for their comment. Through additional experiments we have matched the operating conditions between neutron and x-ray imaged cells to more accurately draw comparisons between the cells.

1. Due to the use of the new datasets with identical discharging conditions the statement quoted by the reviewer is not used anymore and has been changed to the following; line 189: “The neutron images show a similar volume expansion and electrode movement during discharging.”

2. In order to obtain a satisfying sample statistic of the cathode thickness we averaged the electrode height and the electrode length of 4 electrode rings, each ca. 360° . After that the grey values (intensity) were plotted over the electrode width of the four electrodes, see Fig. R 2. The electrode thickness was then measured by using the full wide at half maximum (FWHM) for all four electrode sections. The final result for each of the electrode thicknesses at the different SoC were calculated by the average of the four single values measured for each SoC. The first two steps, the averaging over the electrode height and length, yield a good statistic for local electrode thickness variations. The use of the FWHM for calculating the electrode thickness overcomes

the problem of the relatively large pixel sizes by interpolating the intensity between the measured pixel values. Further, using the average of four electrode sections per SoC minimizes the error of the FWHM measurement. Through these methods we maximized our sample statistics.

3. Thank you for highlighting this, we used two different but commercially identical batteries. For clarity we have added "... two different ..." in line 77. "X-ray and neutron imaging were performed to study the lithium intercalation, morphology changes and degradation processes in two different commercial CR2 lithium/manganese dioxide cells discharged under identical conditions as described in the Methods / Experimental section. In the following sections, the X-ray and neutron imaging analyses are discussed along with the insights achieved from virtual unrolling of the spiral wound electrode architecture."

line 194: Fig 3b line plot. Tomogram 07 is not in the stacked line plot. Just curious, did the switch from cold to thermal neutrons create too much offset for normalization to the same scale as the other line plots?

Due to the different attenuation coefficient of thermal neutrons for the battery materials we decided to not include the line plot for CR2-07 in Fig. 3b to retain consistency. Fig. R 1 shows a replot of Fig. 3b. The graph shows a big difference in the plotted, normalized intensity between the cold neutrons measured tomograms (01 to 06 and 08 and 09) in comparison to 07 (blue). 07 is located between the graphs 06 and 05 and shows a slightly different shape.

Fig. R 1: The graph shows a replot of the in manuscript NCOMMS-19-09249A-Z in Fig. 3b plotted graph. Here we included the line plot of CR2-07 (blue) which depicts the with thermal neutrons measured graph.

In the new manuscript no results are produced from the use of thermal neutrons.

line 194: Fig 3. The x-axis in the line plots is (I think) a radial distance. Would it make more sense to set the origin of this distance to the battery axial center? We note that Fig 5 uses “length” as distance along an electrode.

We are thankful for that comment. The center of the x-axis is set to the center of the battery cell and the x-axis title has been changed for greater clarity, to ‘radial distance’, see Fig. R 2. We hope that this improves the clarity of the figure.

Fig. R 2: Edited Figure 3. from manuscript NCOMMS-19-09249A-Z.

line 219: Let’s consider -045 image and line plots. The dips in the line plot at slices 36-44 at about 160 mm and 190 mm should be visible in the -045 image. I think I can see the changes, but a call out with color rectangles (as done in Fig 2b “regions of strong...”) would be helpful.

The figure has been relabeled with respect to the new data sets and the definitions of the regions have been clarified. We added labels for the ‘inner electrode ending’, the ‘outer electrode ending’ and the regions of ‘strong lithiation/cracking ...’, see Fig. R 3.

Fig. R 3: Overworked Figure 5 from manuscript NCOMMS-19-09249A-Z.

line 228: “result of the better Li-ion conductivity caused by the excess electrolyte at the bottom”. This points out the author’s failure to clearly establish a coordinate origin for the electrode length. I though origin was at the center per the axis triad in Fig 1d. Now, it appears the origin is the outer layer. This is confusing.

Please see next paragraph.

line 250. The point of Fig 7, activity of electrodes at the very center of the battery, is obscured by the confusion of whether center of battery is a 0 mm or 220 mm. Interesting, the comment “late start” is a powerful indication of the complementary use of X-ray and neutron imaging.

To overcome that problem we did a number of changes in the text, added labels in the figures (e.g. Fig. R 3) and redesigned graphs (e.g. Fig. R 2). Now, all text parts using the battery center and the inner electrode end as coordinate origin. Fig 7 is changed as well and shows the inner electrode ending as origin labeled with 0 mm in the graphs, e.g.;

Line 169: “... makes it possible to plot the intensity changes along the length of the electrode assembly from core to case.”

We thank again the reviewer for the very helpful comments and hope to solve the confusing labeling of the figures.

Great work folks!
 Les Butler
 Louisiana State University
 USA

Reviewer #2 (Remarks to the Author):

The manuscript by Ziesche and co-workers reports the temporally and spatially resolved imaging of a primary battery undergoing discharge using a combination of x-ray and neutron tomography. The sensitivities of x-ray to heavier elements and neutron to lighter elements complement each other in measurements, yielding a clearer picture of the structural evolution of various components in a primary battery with rolled electrodes. This study is a nice extension of the previous effort in applying in situ x-ray and neutron measurements to studying battery cells in operando, in particular the virtual unrolling representation of layered structures allows for semi-quantitative analysis of structural variations across the cell and over the time.

As this Communication reports extending the application of a novel measurement methodology developed in the past years to studying batteries in operando, several noticeable deficiencies need to be addressed to elevate the caliber of the study using advanced techniques in major research facilities.

We thank the reviewer for their constructive comments and sincerely appreciate their thorough review of our work.

1. The reported measurements are not real-time in-operando for neutron, and not truly multimodal or simultaneously probing the same location and the same event of the structure evolution toward achieving the goal of synergistic measurements of hard/soft matter. To take advantage of the complementary compositional sensitivity of x-ray and neutron, it is desirable to apply measurements to a single cell, or at least identical cells operating at identical conditions.

We thank the reviewer for this comment and motivated by this we obtained more neutron beamtime to make directly comparable neutron and X-ray measurements to address (1.) real-time operando measurements for neutrons and (2.) operating the cells, which are measured by neutrons and X-rays, under identical conditions.

1. Referring to your comments, we discussed with our collaboration partners from HZB regarding an operando measurement of a fresh CR2 cell during discharging. Limitations due to the lower neutron flux, in comparison to state-of-the-art X-ray beamlines, we decided to propose for beamtime at the new neutron imaging beamline, D50, at ILL (ca. 10 times higher neutron flux compared to the CONRAD beamline at HZB). There we set up a 4D scan by continuously rotating the battery during discharging. We used an electrical slip ring to connect the CR2 cell to the electrochemicalycler and, again, discharged at a constant current of -200 mA such as used in the previous manuscript. This was because of the relatively long exposure time that we could be used for each projection, in comparison to a higher discharge current.

During the discharging process the cell was continuously rotated and projections were collected with ca. 2 s of exposure time. For the reconstruction we used the iterative reconstruction algorithm SIRT because of the higher image quality compared to standard FBP algorithm. **Fig. R 4** and **Fig. R 5** show the discharging curve of the measurements and selected vertical and horizontal orthogonal slices from the reconstructed tomographies at different SoC. A more detailed description of the measurement procedure and image processing can be found in the figure description of **Fig. R 4** and **Fig. R 5**.

The resulting orthogonal slices show a satisfactory image quality for the very fast neutron scans with a total exposure time of about 1,800 s per tomogram. Tomograms were reconstructed after every 90° rotational step. The lithium removal from the anode and intercalation in the Li_xMnO_2 cathode is visible as well as the electrode cracking around the nickel current collector mesh, the electrolyte consumption during the discharge and dry regions where electrolyte wetting is missing. However, the image data shows a high proportion of noise making it unsuitable for electrode unrolling. For this reason, we decided to use in-situ scans for our

publication with identical discharge conditions for the X-ray and neutron data sets. We also decided not to include the 4D neutron scans (Fig. R 4 and Fig. R 5) in the supplementary section of the manuscript. Please see the next paragraph for more details.

2. Owing to the results of the operando measurements, described above, we decided to carry out experiments to capture the same cell design under X-ray and neutron data under identical discharge conditions. We decided to discharge both measured CR2 cells over a commercial 4.7 Ω resistor. The X-ray measurements were measured in-operando under the same imaging conditions like for the previous scan over a 2.75 Ω resistor, that time with 103 tomograms in total. Based on the X-ray results we decided to capture eight different times during the discharging process, where operation of the CR2 cell was interrupted each time to perform in-situ neutron tomography. For a more detailed description of the measurements please see the method / experimental section in the new manuscript.

The entire manuscript has been updated with the new data where operating conditions are identical. Drawing comparison between the new data, the following observations were made;

1. an identical Li_xMnO_2 volume increase during the cell discharge for the X-ray and neutron scans
2. a sinusoidal-shaped intensity variation over the electrode length with a larger period at the outer electrode windings
3. electrode movement in a radial outward pointing direction

Additionally, we found a new feature in the axial upper 2 mm of the electrodes in the neutron data. Resulting out of the fast discharge we found a stronger electrochemical activity there during the first discharge period which is identified by stronger lithium removal, areas of missing electrolyte and a higher volume increase of that region during the first dc phase. We sincerely thank the reviewer for this suggestion as we consider this modification to have substantially improved the quality of the manuscript.

Fig. R 4: 4D neutron tomography scan of the discharging process of a CR2 primary cell from Duracell with a discharge current of -200 mA: The images show vertical orthogonal slices of a continuously discharged CR2 cell during in operando neutron tomography. During the discharging process the battery was permanent rotated with a speed of about 2 rph and continuous collecting projections with an exposure time of 2 s for each. The used detector pixel size was $8 \mu\text{m}$ and the projections were binned 3×3 , to obtain a better SNR, before 3D reconstruction by using the SIRT reconstruction algorithm from the ASTRA toolbox and applying beam hardening correction. One tomography was reconstructed after ca. every 90° of rotation to realise a better temporal resolution. That 4D data show similar behaviour of the active cell parts, which includes the Li_xMnO_2 and Li-electrodes and the electrolyte, as the in the article described neutron in situ scan. Here we also observe electrode cracking during lithium intercalation, electrode swelling and electrolyte consumption inter alia while discharging. The image data were collected at the D50 neutron imaging beamline at ILL.

CR2-40
-879.67 mAh (15834 s)

Fig. R 5: 4D neutron tomography scan of the discharging process of a CR2 primary cell from Duracell with a discharge current from -200 mA: The images show horizontal orthogonal slices of a continuously discharged CR2 cell during in operando neutron tomography. During the discharging process the battery was permanent rotated with a speed of about 2 rph and continuous collecting projections with an exposure time of 2 s for each. The used detector pixel size was 8 μm and the projections were binned 3x3, to obtain a better SNR, before 3D reconstruction by using the SIRT reconstruction algorithm from the ASTRA toolbox and applying beam hardening correction. One tomography was reconstructed after ca. every 90° of rotation to realise a better temporal resolution. That 4D data show similar behaviour of the active cell parts, which includes the Li_xMnO_2 and Li-electrodes and the electrolyte, as the in the article described neutron in situ scan. Here we also observe electrode cracking during lithium intercalation, electrode swelling and electrolyte consumption inter alia while discharging. The image data were collected at the D50 neutron imaging beamline at ILL

2. Thus the comparison and complementarity are compromised with the high c-rate in x-ray measurements and low c-rate in neutron measurements. There are unique limitations and advantages of each technique, overlap of certain operation conditions are needed.

We thank the reviewer to bring that point to our attention. That comment was one of the reasons to repeat our experiments and use identical discharge conditions for our X-ray and neutron measurements. We hope to address that comment with the new data sets and worked out the advantages and limitations of each techniques.

3. The outlier date point using thermal neutron could be addressed with more rigor, and better quantification is possible through normalization using the flux and cross-section ratios of thermal and cold neutrons.

Thank you for the helpful comment. In the new data we measured exclusively with neutrons in the cold neutron spectra.

However, we want to point out the problem of normalizing measurements with neutrons of a different energy spectra. The attenuating coefficients or neutron cross-sections vary strongly between the different battery components, e.g. lithium is a very high neutron absorber whereas hydrogen shows a low absorbing part but a high incoherent scattering proportion and MnO_2 has a lower cross-section at all. It is quite easy to correct images, measured with a different neutron spectra, when the measured objects consist of one or more than one element but with similar cross-sections. Correcting images with multiple cross-sections which vary heavily show difficulties, such as that images can be corrected just for one value which results in wrong results for other elements or using an averaged value results in the wrong corrected images for all elements. In the case of single elements that are well separated locally, image corrections can be done by correcting each element at the local position in the image. However, the discharging process of a battery cell underlay dynamic diffusion processes which vary the local cross-sections constantly, e.g. electrolyte consumption by cracks in the MnO_2 electrode or lithium intercalation in the cathode. Because of these reasons normalization of the thermal measured neutron data will not provide accurate comparable results.

In terms of neutron flux correction, the images are already flux corrected in the preprocessing of the tomography reconstruction.

4. The statistical significance of the data and observation is not sufficiently addressed. The extents of the generalization of the failure mechanism as observed in this study are not assessed.

We thank the reviewer for bringing this to our attention. To address this point, a number of X-ray and neutron measurements are required to satisfy the statistical significance of the measurements. But due to the very valuable time on synchrotron and neutron imaging beamlines, and the requirement of high exposure times for neutrons, we are not able to satisfy that point. However, we do hope to address this by adding that the following in line 267:

“However, despite the behaviour of the two cells investigated here being very similar, we cannot say for certain that their behaviour is statistically significant and representative of all cells operated under similar conditions.”

5. It is necessary for better quantification of experimental measurements to be more relevant to the electrochemical characteristics of batteries. The comparison between the amount of transported Li and the state of the charge and the correlation between the volume change and the change of Li content could be addressed.

We thank the reviewer for this constructive feedback. The comparison between the transported amount of lithium, the SoC and the correlation between the volume change in comparison with intercalated amount of lithium is extremely challenging:

There are a number of things to address which impede that quantification; Firstly, the distinction between the remaining lithium and electrolyte is very challenging, if at all possible for this kind of experiment, due to the similar attenuation coefficient of the hydrogen in the electrolyte and lithium. Secondly, the Li_xMnO_2 electrode undergoes cracking during the discharging which complicates the distinction between active cathode material and voids which are partially not visible in the neutron images due to the filling up by electrolyte. Furthermore, to remove the nickel current collector mesh out of the neutron data is possible just with advanced segmentation tools like a Weka filter and even by using such a filter the segmentation shows a large error. A scenario where this may be possible could be where the exact same cell is imaged by both neutron and X-ray, and the data overlaid. Here it is possible to mask voids and the current collector mesh using the X-ray results and apply to the neutron data. This would make a useful study, however, would be a large time commitment beyond the scope of this work.

To bring that difficulty to attention we added the following in line 302 : “Due to the electrolyte and lithium having similar attenuation coefficients for neutrons, a direct correlation between the SOC and the amount of lithium consumed could not be made, highlighting a limitation of this technique for the most commercially relevant cells.”

6. In general, the focus of the study is mostly on engineering aspects of primary batteries. It is desirable to apply novel measurement techniques to the state of the art battery materials or new battery architectures and addressing the most relevant scientific issues in advancing battery technology.

We agree with the reviewer that great benefit is to be gained from applying such novel measurement techniques on state-of-the-art battery materials, such as Si containing 18650 Li-ion cells. The experimental procedure that we outline here may provide a framework for future studies on cutting edge materials, however, before larger geometries (e.g. 18650s, 21700s) may be analyzed in such a way, improvements in probe-flux are required, however, the framework and analysis algorithms remains the same.

Here the main reason for the low transmission is the high amount of strong neutron attenuating materials such as Co and Mn beside the Li and H. We are confident to overcome that issue by using the new D50 high flux imaging beamline at ILL or later the planned new imaging beamline Odin at ESS.

This comment outlined an important aspect of the work, consequently we have added the following amendment to the manuscript in order to highlight the potential future applications, line 305:

“We expect that this methodology will be valuable for studying and optimizing a range of battery chemistries, not limited to the primary cell studied here it will also make great benefits in the field of rechargeable cells.”

Recommendation -- the manuscript could be further improved. The scheme and outcome could fit to a journal more specialized on methodology or toward electrochemical research community.

Reviewer #3 (Remarks to the Author):

This is an interesting paper detailing a novel combination of imaging methods to extract insightful information about the battery discharge process. This kind of work is an important contribution towards the goal of understanding the connections between manufacturing/production methods and discharge performance. The method gives significant insight into transient, spatially resolved behaviors relevant to battery performance. While not done here, using these techniques to study transient discharge behavior in different battery designs could offer impactful new insights as to why some designs or manufacturing methods result in superior capacity/cycle life, which in many cases can currently only be hypothesized with the help of end-state battery imaging or computational modeling. I suggest some minor revisions before publication:

We thank the reviewer for their helpful suggestions and would like to inform that we are expanding the developed techniques to other relevant battery types and chemistries.

1. Lines 178-182:

a. Can you be sure that the differences in electrode expansion between medium and high rate discharges are due to cracking and not due to differences in imaging technique and/or image postprocessing? Is resolution sufficient enough to confidently report these volume expansion numbers without more statistics? Even a +-reporting of spread throughout the length of the electrode to accompany the average value reported would give these measurements more weight. It may be stated elsewhere, but how does the pixel size compare to the volume expansion: how much error is introduced if one pixel of expansion is incorrectly included/not included in the expansion measurement?

We are sure that all differences in electrode expansion of the X-ray and neutron data are the result of the cracking and lithium intercalation process. As a reference we measured the thickness variation of the cell casing which showed a very constant value over the single SoC. Please find as reference the thickness variation

of the battery casing for the X-ray tomographies in **Fig. R 6** below and for the neutron tomograms in **Fig. R 7** below for 3 tomograms of the measured data respectively. For that error calculation we used the intensity variation of the cell casing over the discharging process. As reference the intensity profile of a piece of the casing out of a vertical orthogonal slice was used and for a better statistic the profile was integrated over about 100 pixel lines. The thickness was then determined by using the FWHM of the intensity plot for the X-ray and neutron data respectively. For the X-ray and neutron data a standard deviation of $< 1.5 \mu\text{m}$ over the discharging was calculated. The averaged casing thickness, over the discharge, was determined to be $175.5 \mu\text{m}$ and $174.6 \mu\text{m}$ for the X-ray and neutron data, respectively. The minor difference in the averaged thickness can be traced back to a slight rotation in the vertical cell position.

Due to the fact, that the here calculated error shows a smaller deviation as $1.5 \mu\text{m}$ we decided not to include an error bar in the images.

Furthermore, in order to obtain a satisfying sample statistic of the cathode thickness we averaged the electrode height and the electrode length of 4 electrode rings, each ca. 360° . After that, the grey values (intensity) were plotted over the electrode width of the four electrode, see **Fig. R 2**. The electrode thickness was then measured by using the full width at half maximum (FWHM) for all four electrode sections. The final result for each of the electrode thicknesses at the different SoC were calculated by the average of the four single values measured for each SOC. The first two steps, the averaging over the electrode height and length, yield a good statistic for local electrode thickness variations. The use of the FWHM for calculating the electrode thickness overcome the problem of the relatively big pixel sizes by interpolating the intensity between the measured pixel values. Further, using the average of four electrode sections per SoC minimizes the error of the FWHM measurement. In that way we did the best to achieve a high level of sample statistics.

Due to the high interest in the error of that techniques we decided to include **Fig. R 6** and **Fig. R 7** as Figure S 7 and S 8 in the supplementary information section of the manuscript.

Fig. R 6: Error calculation by using the intensity variation of the cell casing during the discharging process. As reference the intensity profile of a piece of the casing out of a vertical orthogonal slice was used and for a better statistic the profile was integrated over about 100 pixel lines. The thickness was then determined by using the FWHM of the intensity plot. Here shown for the X-ray data the calculated error between the measurements show a deviation smaller as $1.5 \mu\text{m}$.

Fig. R 7: Error calculation by using the intensity variation of the cell casing during the discharging process. As reference the intensity profile of a piece of the casing out of a vertical orthogonal slice was used and for a better statistic the profile was integrated over about 100 pixel lines. The thickness was then determined by using the FWHM of the intensity plot. Here shown for the neutron data the calculated error between the measurements show a deviation smaller as 1.5 μm.

b. What is the theoretical/measured expansion of pure LiMnO_2 upon lithiation and how does that compare to these measured values? This may give insight into how much porosity (and therefore Li^+ transport through the pore network) is decreasing due to swelling, at least in the medium C-rate case where there is less cracking.

That's a very interesting point. In the following we try to compare the theoretical volume change of Ramsdellite- MnO_2 (R-MnO_2), the main component of the active MnO_2 material, with the measured volume expansion of the under different discharge conditions cycled CR2 battery Li_xMnO_2 electrodes. But at first, we like to outline that we are not able to find relevant data which describe the volume change of Li_xMnO_2 cathode material by using electrochemical cycling versus a lithium electrode and using electrolyte as Li -ion transporter. This data using the volume expansion of the R-MnO_2 unit cell as reference ¹ for the volume expansion of the pure Li_xMnO_2 electrode. **Tab. R 1** shows the used reference values of the volume increase of the R-MnO_2 unit cell for different lithiation states. We have to mention that the values used here will not be accurate in case for electrochemical intercalated MnO_2 by using electrolyte.

Fig. R 8 shows the calculated percentual volume expansion of the Li_xMnO_2 electrode for four different CR2 cells discharged under different conditions compared with the theoretical volume expansion of the R-MnO_2 unit cell. The cathode volume is corrected by removing the constant volume of the nickel current collector mesh from the total electrode volume (about 8 % in the pristine state). For that calculation we need to assume that the pristine cathode consists of pure MnO_2 with $x = 0.0$ and the state where the cell reaches the maximal capacity of 800 mAh the cathode reaches the maximal lithiation of the Li_xMnO_2 active material with $x = 1.0$. That assumption is just a rough estimation because at small discharge currents the CR2 cell shows a higher capacity as written in the data sheet. To achieve a more precise solution additional XRD measurements will be helpful to determine the correct lithiation states of the electrode. However, the plot shows a good agreement between the theoretical Li_xMnO_2 volume expansion (red line) and the volume expansion of the slowest discharge condition of -200 mA (black line and blue square). With increasing discharge speed (smaller used resistors) the volume expansion increases. As mentioned in the manuscript, the larger volume expansion at fast discharges can be explained by the more pronounced cracking of the electrode. Here a comparison between the Li_xMnO_2 and the void volumes can be helpful but is just possible for the X-ray data.

Due to the made assumptions which are necessary to get the data below, we would prefer not include that calculation in our manuscript. However, we are thankful for the very interesting discussion and will consider including such a study in a future work.

Tab. R 1: Volume expansion of Li_xMnO_2 during lithiation in percentage.¹

State of Lithiation	Volume expansion [%]
$\text{Li}_{0.0}\text{MnO}_2$	0.0
$\text{Li}_{0.3}\text{MnO}_2$	11.1
$\text{Li}_{0.5}\text{MnO}_2$	13.7
$\text{Li}_{0.9}\text{MnO}_2$	21.5

Fig. R 8: The Li_xMnO_2 volume expansion during lithiation x as a comparison between theoretical expansion and expansion under different discharge conditions.

c. You chose to report averaged volume increase, yet for the other metrics in subsequent figures you show the values as a function of length around the spiral cathode. Can you state why? Is the volume increase uniform around the cell or was there sinusoidal behavior like you see in Figure 6? For example, do you see more significant electrode expansion opposite the tab current collector to support the conclusion in line 214/215 and 218?

The data show a number of different features which needed to be addressed. Fig 3 shows the volume change of the whole Li_xMnO_2 electrode averaged over the electrode height and length. Here we get no information about local variations of the volume change. The subsequent figures, Fig. 5 and 6, show the intensity change of

the cathode over the electrode length during lithium intercalation. That is not a measure of electrode volume expansion itself, but is strongly correlated. The sinusoidal intensity change is related with the volume expansion. Thus, for the X-rays, an intensity decrease (line 224: "... It can be concluded that at this side, the compression of the active material is less, and favored an easier intercalation/cracking followed by a simultaneous volume expansion.") and for the neutron an intensity increase (line 251: "The local intensity increase is also strong related to a local volume increase of the cathode.") marks a higher electrode volume expansion. To make that clearer we added the highlighted sentences to the manuscript for a better understanding.

In regard to line 214/215 we can see a more pronounced volume expansion at the opposite tab side for the over 2.75 Ω discharged X-ray data. In tackling comments from the other reviewers, we carried out a new set of experiments where the same operating conditions were duplicated for the neutron and X-ray imaging experiment, thus facilitating direct comparisons. For the new X-ray data where the cell was discharged over a 4.5 Ω resistor we have a higher volume increase at the outer electrode windings, see line 217; "A higher intensity change is clearly visible at the outer windings of the electrode (left side from the current collector), with a decrease observed over the electrode length inwards, which indicates higher lithium intercalation or cracking caused by a lower compression of the active material."

2. Line 190-191: Since this statement is repeated several times throughout the paper during the analysis, it should be more detailed by explaining why this statement is true (A lower compression results in a more porous electrode, allowing for easier lithium diffusion through the pore network as well as providing more room for mesoscale electrode expansion into the pores as well as macroscale expansion via cracking of a less dense/strong composite material.) Up to this point in the paper, there has been no mention of electrode porosity, so it may not be clear why the statement is true.

We thank the reviewer for bringing that to our attention. Due to that question we checked the manuscript where the electrode compression and porosity is mentioned for the first time and improved the understanding of that section. We hope to have addressed the reviewers comment with the following modification:

Line 102: "Stronger Li_xMnO_2 cracking is detected in the upper axial cathode section than in the middle section, likely as a result of the weaker electrode compression which facilitates lithium intercalation and electrode expansion (see Figure 2 a), Figure S3 and Figure S4). Delamination of active material from the current collector can create electrically (and therefore electrochemically) isolated areas which, in turn, can cause power and capacity loss. However, cracks exposing additional Li_xMnO_2 surface area by increasing the electrode porosity which may also benefit lithium insertion and cell performance due to voids which are fluted by highly Li-ion conductive electrolyte."

3. Line 249. You mention mechanisms, but only state one mechanism. Lithium may not necessarily have to diffuse through the cathode material to get to the cathode endings. Early in discharge, there may be no incentive for the lithium to transport out to the cathode endings through the electrolyte given that there is much closer cathode material available. As the state of charge of the nearby cathode material increases and approaches capacity, the longer transport through the electrolyte to the cathode endings (low SOC) likely becomes the most favorable path. I would think a longer transport path through electrolyte to low-SOC cathode is the likely mechanism here, rather than lithium diffusion through the active material as you state.

We thank the reviewer for their constructive comment. We agree with their suggestion and added the statement to the manuscript as follows:

Line 264: "Consequently, the most likely mechanisms for the lithium insertion in the electrode endings is lithium diffusion through the Li_xMnO_2 electrode or that a longer transport through the remaining electrolyte becomes more likely at a lower SoC, assumed the endings are sufficient wetted by electrolyte."

4. While not directly related to this work, the impact on computational battery modeling could be mentioned somewhere in the introduction/conclusions. The transient, 3D spatially-resolved image data of lithiation and electrode swelling presented here would be invaluable to the modeling field that is quite starved for such detailed model validation data. For transient discharge simulations, typically only global measurements are available (discharge curves, macroscale pouch swelling, etc.).

The authors certainly agree. The following passage has been added to the text line 313:

“Furthermore, that data presents an example of the powerful insight that may be gained from 4D imaging; the bulk properties that computational models were previously based upon, may now be replaced by spatially resolved, transient values. The implications of this lie in that models would not need to apply the macroscopic assumptions made by averaging bulk properties and consequently, due to the capability of incorporating local heterogeneities, they would be able to provide a more accurate and comprehensive description of the practical operation, degradation and failure of a cell.”

References:

1. Thackeray, M. M. *et al.* Ramsdellite-MnO₂ for lithium batteries: the ramsdellite to spinel transformation. *Electrochim. Acta* **38**, 1259–1267 (1993).

REVIEWERS' COMMENTS:

Reviewer #1 (Remarks to the Author):

The revised manuscript is excellent and ready for publication. This is a truly excellent work and will be studied intensely by battery and imaging researchers for years. Great work!!!

Reviewer #2 (Remarks to the Author):

The revision has demonstrated much improved technicality and consistency. This is an important contribution to the very active field of the battery research. However, given the lack of pressing scientific case, and the technique-oriented presentation style, I consider this work most attractive to specialists.

Reviewer #3 (Remarks to the Author):

All of my concerns have been sufficiently addressed by the revision, and the paper is improved due to the additions made by the authors.

I will mention that some of the added discussions within the paper show grammatical errors or awkward wording issues, such as lines:

108-110

192-194

313

443

I suggest that the authors carefully edit/review the added text before final publication, but do not think this requires another round of reviews.

Response to Reviewers' comments:

Comment:

We greatly appreciate the reviewers time on this manuscript and extend our thanks for the constructive comments.

Reviewer #1 (Remarks to the Author):

The revised manuscript is excellent and ready for publication. This is a truly excellent work and will be studied intensely by battery and imaging researchers for years. Great work!!!

We thank the reviewer for his work and the helpful comments.

Reviewer #2 (Remarks to the Author):

The revision has demonstrated much improved technicality and consistency. This is an important contribution to the very active field of the battery research. However, given the lack of pressing scientific case, and the technique-oriented presentation style, I consider this work most attractive to specialists.

We thank the reviewer for their constructive comments and sincerely appreciate their thorough review of our work.

Reviewer #3 (Remarks to the Author):

All of my concerns have been sufficiently addressed by the revision, and the paper is improved due to the additions made by the authors.

I will mention that some of the added discussions within the paper show grammatical errors or awkward wording issues, such as lines:

108-110

192-194

313

443

I suggest that the authors carefully edit/review the added text before final publication, but do not think this requires another round of reviews.

We thank the reviewer for their constructive comment. In the final version we changed all grammatical issues and prove read the manuscript carefully. Many thanks for the spended time and the helpful suggestions.